# Real-Time Bucket Pose Estimation Based on Deep Neural Network and Registration Using Onboard 3D Sensor

**DOI:** 10.3390/s23156958

**Published:** 2023-08-05

**Authors:** Zijing Xu, Lin Bi, Ziyu Zhao

**Affiliations:** 1School of Resources And Safety Engineering, Central South University, Changsha 410083, China; 215512093@csu.edu.cn (Z.X.); 225501023@csu.edu.cn (Z.Z.); 2Digital Mine Research Center, Central South University, Changsha 410083, China

**Keywords:** 3D point cloud, pose estimation, semantic segmentation, real-time

## Abstract

Real-time and accurate bucket pose estimation plays a vital role in improving the intelligence level of mining excavators, as the bucket is a crucial component of the excavator. Existing methods for bucket pose estimation are realized by installing multiple non-visual sensors. However, these sensors suffer from cumulative errors caused by loose connections and short service lives caused by strong vibrations. In this paper, we propose a method for bucket pose estimation based on deep neural network and registration to solve the large registration error problem caused by occlusion. Specifically, we optimize the Point Transformer network for bucket point cloud semantic segmentation, significantly improving the segmentation accuracy. We employ point cloud preprocessing and continuous frame registration to reduce the registration distance and accelerate the Fast Iterative Closest Point algorithm, enabling real-time pose estimation. By achieving precise semantic segmentation and faster registration, we effectively address the problem of intermittent pose estimation caused by occlusion. We collected our own dataset for training and testing, and the experimental results are compared with other relevant studies, validating the accuracy and effectiveness of the proposed method.

## 1. Introduction

In recent years, with the growing demand for resources and the increasing emphasis on the safety of mining operations, the development of mining is moving towards intelligence, with the ultimate goal of achieving autonomous mining. The technical conditions and environmental characteristics of open-pit mine are relatively unique compared to underground mining, and open-pit mines have the advantages of an open transport space, fixed driving routes, low speed and high-quality communication. For these reasons, open pit mines have become typical scenery for intelligent mining construction [1]. A traditional open pit mine site is shown in Figure 1, where the mining operations are carried out by a combination of bucket operators and site supervisors. This mode of excavation is low in efficiency and high in labor costs. Automatic mining [2] and safety assessments [3] can be realized by installing sensors such as LiDAR, camera, and GPS on the mining excavator. This is beneficial to improve the productivity of mining operations [4] and enhance the intelligence of the mining excavator [5,6]. As a key component and end-effector of the mining excavator, bucket pose estimation is an important way to improve the intelligence of the mining excavator [7]. On the one hand, the bucket pose estimation can assist the bucket driver in excavation and loading [8], improving the efficiency of mining operations. Real-time bucket pose estimation also provides an important basis for the realization of automatic mining and loading [9], which greatly contributes to promoting the intelligence of mining [10].

The traditional method for estimating bucket pose is installing multiple non-visual sensors, such as a potentiometer, rotary encoder, and resolver [11,12,13]. To accurately obtain the changing poses of the bucket when mining, several different types of non-visual sensors need to be installed. However, these suffer from cumulative errors caused by loose connections and a short service life caused by strong vibrations.

To solve these problems, we propose a visual-sensor-based bucket pose estimation method. The key contributions of this work are as follows:

To adapt the point transformer model for the bucket point cloud semantic segmentation task, we adjust the parameters of the original model. This adjustment significantly enhances the segmentation accuracy and overall model performance.

By employing pose canonicalization, we avoid the issue of limited overlap between consecutive frames’ bucket point clouds, which can negatively impact registration accuracy. As a result, we effectively address the challenge of achieving accurate bucket pose estimation when the bucket is occluded.

The proposed method does not rely on prior models for registration, making it highly generalizable. It can be applied to estimatations of the pose of end-effectors of other pieces of loading equipment in an open-pit mine.

## 2. Related Work

In this section, we will comprehensively review the relevant studies and technological advancements pertaining to pose estimation.

Considering the problems of non-visual sensor measurement methods, some researchers adopted visual sensor methods to estimate the bucket pose. Wen [14] present a monocular vision-based method for 3D excavator pose estimation, which can provide the groundwork for monitoring contact-driven accidents near excavators. Mahmood [15] proposed the use of synthetic images for training a CNN-based 3D pose estimation model, which was tested and applied to actual images from job sites. However, image-based pose estimation methods suffer from significant errors, and they are unable to estimate the bucket pose when it is occluded during mining operations.

To enhance the accuracy of pose estimation, Tang [16] proposed a multiple-onboard-sensors data fusion approach for full-body pose estimation of an excavator. By combining the advantages of IMU and camera sensors, they improved the accuracy of pose estimation. This method outperformed methods relying solely on IMU or camera for state estimation. However, it still exhibited a relatively large average error of 9.066 cm. Cui [17] proposed a novel memory augmented registration network for bucket pose estimation. This kind of method can effectively alleviate the forgetting problem and increase the estimation accuracy of the bucket poses with small samples. This method demonstrated relatively few errors (a rotation error of 2.141 and translation error of 4.8 cm). However, it still faces the challenge of requiring a large number of samples for training. Additionally, the experiments were conducted on empty buckets without any occlusion, and the accuracy of pose estimation for occluded buckets was not validated.

Therefore, in practical applications of mining operations, there is a need to explore a pose estimation method that offers high accuracy and is not affected by occlusion.

Recently, the combination of registration and deep neural networks has achieved a remarkable performance in pose estimation [18,19,20,21]. Wang et al. [22] utilized the modified U-SegNet segmentation network to segment the target and employed iterative closest point (ICP) [23] to register the segmented results with the pre-scanned model candidates for pose estimation. Zeng et al. [24] applied a fully convolutional network (FCN) [25] to segment and label multiple views of the scene. They used ICP to align the segmented point clouds with the pre-scanned model to estimate the 6D pose of objects. Similarly, V. Tran et al. [26] optimized the registration between target objects and CAD models based on R-CNN and structured light technology. Yang et al. [27] improved the pose estimation performance in partially occluded scenarios by combining object detection networks and ICP registration in a life-support robot operating in real-world environments. Luan et al. [28] proposed a multi-path convolutional neural network BiLuNet, which is based on the U-Net framework. The network extracts the object mask and merges it with the depth information to perform 6D pose estimation using the Iterative Closest Point (ICP) algorithm. Compared to the U-Net with limited training data, their method achieved better results.

These methods are all based on the use of pre-scanned models for registration, which leads to slower registration speeds. For example, Luan’s method took 2.76 s to process [28], and Wang’s method had a total processing time of from 2 to 3 s from sensor data capture to pose estimation [22]. These processing times do not meet the real-time requirements of loading scenarios. Moreover, these methods have not been proven to work in environments with significant occlusion. Therefore, before applying these methods in open-pit mine scenarios with a high level of interference such as dust, noise, and pile collapse, it is necessary to address the challenges of severe occlusion and real-time computation.

We apply a method that combines registration and a deep neural network in the mining field. By employing point cloud Pose Canonicalization, we effectively address the challenge of difficult registration when the bucket is occluded. Additionally, we accelerate the registration speed of the FICP algorithm by removing outlier points and downsampling the point cloud, ensuring real-time performance. This allows for continuous pose estimation of the bucket with occlusion in complex open-pit mine scenarios.

## 3. Methods

Our method is summarized in Figure 2, with four main steps: model training, point cloud segmentation, point cloud processing, and pose estimation. After receiving a sequence of point clouds as input, we first utilize a model for semantic segmentation to classify different objects in the point cloud. The segmentation results are subjected to outlier removal and downsampling. Next, the point cloud is pose-canonicalized and registered with the first frame’s point cloud using the FICP registration algorithm. Finally, the current frame’s pose estimation is computed based on the registration result.

### 3.1. Semantic Segmentation

The main types of point-cloud-processing task include object detection and semantic segmentation. In the source data of this research, the bucket and pile are involved in a dynamic interaction process. It is challenging to accurately separate the bucket and pile based solely on candidate bounding boxes generated by object detection. Therefore, semantic segmentation is employed, treating individual points as the unit of judgment. Points that do not belong to the bucket class are directly removed from the point clouds based on point labels, enabling the most precise bucket segmentation and ensuring accurate pose estimation.

Engel [29] proposed a Point Transformer (PT) network that applies self-attention to 3D point clouds. The PT network exhibits strong scalability, superior performance, and high segmentation accuracy, making it suitable for bucket segmentation tasks. It can achieve the desired segmentation performance even with relatively few training data and in the presence of occlusions.

To make the PT network more suitable for the bucket segmentation task and improve model performance, we adjusted parameters based on the original model. Considering the sparse density of point clouds in bucket loading scenes, we moderately reduced the number of groups used for attention calculation and set the number of nearest neighbors in the neighborhood for local attention computation to 8. This could enhance the model’s ability to capture spatial relationships and semantic information in bucket loading point clouds. We also accelerated the training and convergence speed by reducing the number of network layers and feature vector dimensions while ensuring model performance.

Since the FICP algorithm is used for registration, the registration accuracy can be affected by outliers. Therefore, when applying the model to our bucket point cloud segmentation task, we focused more on segmentation accuracy. We set the dropout rate parameter to 0.5 to enhance the model’s robustness and achieve high-precision segmentation. The model’s performance evaluation metric for bucket point cloud semantic segmentation is the mean Intersection over Union (mIoU). The optimal parameter combination for the model in the bucket segmentation task is presented in Table 1. Detailed experiments related to parameters adjustments are described in Chapter 4.

### 3.2. Pose Estimation

Achieving accurate bucket pose estimation is crucial for practical applications such as autonomous mining and loading. In addition to ensuring the accuracy of semantic segmentation results, the accuracy of the registration algorithm is also crucial. However, the bucket is often occluded by pile collapse and dust during the mining operations, which makes registration extremely challenging. Therefore, in this paper, we used the Fast Iterative Closest Point (FICP) [30] algorithm for registration. Compared to the classical ICP algorithm, FICP uses a robust error metric based on the Welsch function, which reduces misalignment in the presence of outliers and partial overlap.

Although the FICP method has made significant improvements compared to the ICP algorithm, there are still some issues. Despite the use of Anderson acceleration [31] to improve convergence speed, FICP is still sensitive to the initial relative pose, making it prone to local optima and meaning that it has a slower computation speed when the distance between the two frames is not close enough for registration. To address the sensitivity of the FICP algorithm to the initial relative pose, this paper adopts continuous frame registration for tracking. Due to the temporal continuity, the bucket’s pose changes minimally between consecutive frames, and the distance between the bucket point clouds is extremely close. This approach ensures fast registration using the FICP algorithm while avoiding getting trapped in local optima.

We can represent a 6DoF transformation T={R,t}. Since more of the bucket point cloud is lost when there is occlusion, there is little similarity between the bucket point clouds of consecutive frames, which align poorly across different frames. Inspired by [32], we turned the estimation of $T_{t+1}$ into ${\overset{\wedge }{T}}_{t+1}$, which is widely used in instance-level 6D pose estimation and tracking works [33,34].

The input point cloud $X_{t+1}$ was pose-canonicalized for each frame except the first before alignment. The pose-canonicalized point cloud $Z_{t+1}$ is defined as the product of the inverse transformation of $X_{t+1}$ and $T_{t}$.

Due to the continuity of the pose between frames, the pose-canonicalized point cloud is very close to the first frame point cloud, and this step could be considered the initial alignment. Note that $T_{t+1}$ can be expressed using ${\overset{\wedge }{T}}_{t+1}$ and $T_{t}$, namely, $t_{t+1}=R_{t}{\overset{\wedge }{t}}_{t+1}+t_{t}$, $R_{t+1}=R_{t}{\overset{\wedge }{R}}_{t+1}$.

In this way, we can reduce the effect of bucket point cloud occlusion on pose estimation. By doing so, we significantly simplify the regression task and ensure accurate pose estimation even when part of the bucket point clouds is missing.

## 4. Experiment

A series of experiments were conducted in both a CARLA simulation environment and a metal mine to evaluate the accuracy and effectiveness of the proposed method. The experiments consisted of two main components: semantic segmentation and pose estimation. This section also introduces a new dataset that was used to train the network and validate the performance of the proposed method.

### 4.1. Dataset

The existing large-scale point cloud datasets, such as S3DIS, do not include data based on mining environments. Therefore, based on a simulation environment and a metal mine, we collected a new dataset specifically for bucket pose estimation.

We built a simulated mining scene based on a specific mine in the simulation platform CARLA. CARLA is an open-source research simulator for autonomous driving, based on the Unreal Engine 4 (UE4), which provides good simulation effects for real-world scenarios and physical activities.

In the simulated scene, the bucket is raised to a certain height and tilted backward to deposit the material into the transport vehicle after excavation from the pile. Then, the bucket returns to the excavation position to repeat the loading process. A LiDAR sensor installed on the driver’s cabin of the excavator synchronously collects environmental point clouds at a frequency of 10 Hz. The data collection duration is 3 min (one cycle of loading operation). We selected 332 frames containing the point cloud of the bucket to form the simulated dataset (Figure 3).

In the real metal mine, real mining and loading data were collected using a fixed-point acquisition scheme. The LiDAR sensor was fixed on the excavator’s cabin to capture the environmental point clouds during the excavator’s mining and loading operations. The total data collection duration was 2 min, from which 1158 frames containing the bucket point cloud were extracted. Due to the repetitive nature of the mining and loading process, a final selection of 100 frames was established as the real-world dataset.

### 4.2. Semantic Segmentation

To improve segmentation accuracy, we explored the effects of four parameters on model performance. The goal was to identify the best combination of PT parameters for bucket point cloud segmentation. For comparison purposes, an original model with unmodified parameters was set up. Figure 4 provides insights into how adjusting various parameters of the PT model affects its performance, using mIoU as the evaluation metric. The model parameter configurations are shown in Table 2.

Observing graph (a), it can be seen that reducing the number of neighboring points leads to better model training results compared to the original model. Since open-pit mine environments are more spacious, the number of point clouds collected by LiDAR is significantly lower than that in indoor and other densely populated areas. Therefore, a reduced number of neighboring points is sufficient to capture the spatial structure of most objects. Models with fewer neighboring points achieve a better performance as there is less interference and noise.

From graph (b), it is evident that when the number of groups is set too large, the final convergence results are lower compared to both the original model and the model with a reduced number of groups. Increasing the number of groups potentially improves the model’s ability to adjust its features while retaining error information during the computation process. For the bucket segmentation task, individual points may not require fine-grained feature representations but are more sensitive to error information. The improved performance of the model with reduced group numbers validates this analysis.

Graph © illustrates the impact of dropout rates of 0.25 and 0.5 on model performance. The experiment reveals that the highest dropout rate also achieves the highest segmentation accuracy, indicating that the model’s resistance to interference is indeed enhanced by an increase in the dropout rate.

Analyzing the results in graph (d), it can be observed that decreasing the number of network layers improves initial performance and speeds up convergence. On the contrary, increasing the number of network layers leads to more significant oscillation. Ultimately, the performance of models with increased and decreased network layers does not significantly differ from the original model, indicating that adjustments to the number of network layers are unnecessary.

Based on a comprehensive comparison, the best-performing parameter combination is selected for the PT model, focusing on its segmentation performance on the simulated dataset. The selected parameter combination is then used for model retraining.

We trained several different models using the same dataset, including Sparse-Unet, KP-FCNN, and the original PT model. The mean Intersection over Union (mIoU) curves of the training process are shown in Figure 5, with the mIoU curve of the parameter-optimized PT model represented by a bold red line. From the graph, it can be observed that both the original PT model and the parameter-optimized PT model significantly outperform the other two models. Furthermore, the mIoU curve of the parameter-optimized PT model exhibits reduced fluctuations and faster convergence compared to the original PT model.

We tested several different segmentation models, including PT, original PT, and Sparse-Unet, with the same test set including 55 frames of the simulated dataset and 55 frames of the real-world dataset. The purpose was to validate the effectiveness and accuracy of the segmentation models. In this experiment, the evaluation metrics include Overall Accuracy (OA), mean Intersection over Union (mIoU), mean Accuracy (mA), and Intersection over Union (IoU) for each class. The results for the simulated dataset and real-world dataset are presented in Table 3 and Table 4.

Our optimized PT model demonstrated an excellent performance on the simulated dataset, achieving an average IoU of 98.16% and an IoU of 96.05% for the bucket class. It outperformed the other two models across all evaluation metrics. On the real-world dataset, where the models were exposed to disturbances such as dust and noise, their performance declined. However, our method still achieved accurate segmentation results. The average IoU for the real-world dataset was 88.53%, with an IoU of 87.03% for the bucket class. While Sparse-Unet had a higher OA of 96.45% and a higher mA of 94.33%, its IoU for the bucket class was only 82.90%, with an accuracy of 86.02% for the bucket class, which was significantly lower than the PT model’s IoU of 87.03% and accuracy of 90.36% for the same class. These results demonstrate that the parameter-optimized PT model outperforms other architectures in terms of accuracy and IoU.

Unlike the simulated environment, real scenes present with more interference factors. In addition to bucket occlusion caused by the blast pile, dust near the bucket also introduces disturbances to semantic segmentation. As a result, the segmentation accuracy of the real scene dataset is slightly lower compared to the simulated environment dataset. However, it still achieves satisfactory segmentation results.

A visualized comparison of the output segmented results is shown in Figure 6. This compares the segmentation results of different models with the ground truth point clouds. From the comparison, it can be observed that the parameter-optimized PT model exhibits segmentation results that closely resemble the ground truth. The boundaries between the red, blue, and green point cloud classes are clearly visible, and the segmented bucket points closely match the ground-truth annotations.

In contrast, in the © group of images depicting three different bucket poses, the segmentation results of the red, blue, and green point cloud classes are poorer. Both the bucket and boom class segmentation results include outliers from the surrounding environment, which are included in the segmented point cloud and disrupt the registration process. In the third pose, due to the influence of surrounding dust, only a partial segment of the bucket is obtained.

In summary, as the degree of occlusion increases, the accuracy of the Sparse-Unet model’s segmentation decreases. It fails to accurately segment the surrounding environmental elements, such as dust and blast piles, along with the bucket. On the contrary, the PT model demonstrates remarkable accuracy, precisely segmenting objects of all three categories from cluttered point clouds. It effectively segments as many points as possible within the bucket class, preserving the shape characteristics of the bucket. This is crucial for the precise registration in subsequent frames.

The accuracy of semantic segmentation affects pose estimation. In our approach, we tested the semantic segmentation model on our own dataset, and the results demonstrate that our segmentation model achieves precise segmentation.

### 4.3. Pose Estimation

In this experiment, due to the difficulties of calibration in real-world scenarios, it is challenging to obtain accurate bucket pose estimation for quantitative analysis. Therefore, we selected a continuous sequence of 20 frames of point clouds from the simulated dataset for experimentation. To minimize the potential impact of outliers on registration accuracy and improve registration speed, the segmented point clouds are preprocessed before registration. Points located 25 m from the lidar are removed based on the actual range of bucket motion. After removing outliers and downsampling, we reduce the point cloud size from 3000 points to 1500 points. This significantly accelerates the registration speed of the FICP algorithm.

Table 5 presents the rotation error, translation error, and registration time for bucket pose estimation using various classic registration algorithms, including ICP, SAC-IA, RANSAC, Sift + RANSAC, GICP, and FICP. ICP performs the worst due to its susceptibility to local optima. SAC-IA and RANSAC, two feature-based matching algorithms, demonstrate improved accuracy compared to ICP, with RANSAC showing a slower computation speed. After incorporating the SIFT keypoint extraction step, the RANSAC algorithm significantly improves both registration time and accuracy. Variants in ICP, GICP, and FICP show substantial improvements in terms of both accuracy and time, especially FICP. FICP achieves exceptional registration accuracy with an average rotation error of 1.21° and translation error of 2.51 cm, meeting the high registration precision requirements of the application. While GICP exhibits a faster registration time of 20 ms compared to FICP’s 75 ms, FICP provides a higher registration accuracy. Moreover, considering the experimental use of a lidar frequency of 10 Hz, with one frame of point cloud data being transmitted every 100 ms, the real-time requirements of the application are satisfied by the FICP algorithm.

Figure 7 compares the registration results of different registration algorithms. It clearly demonstrates that FICP outperforms other registration algorithms in terms of registration accuracy. It can also be observed that ICP performs the worst among these registration algorithms, as indicated by the red point cloud, which becomes increasingly distant from the blue point cloud as the occlusion of the bucket increases. SAC-IA and RANSAC algorithms have a relatively better performance, with the red and blue point clouds appearing closer in poses 1, 2, and 3. However, in pose 4, the distance between the red and blue point clouds becomes larger. In contrast, FICP demonstrates a consistently good performance across all poses, with the red and blue point clouds almost overlapping.

Overall, the ICP algorithm exhibits significant registration errors between the transformed point cloud and the target registration point cloud. Moreover, as the degree of the occlusion of the bucket increases, the registration error also increases. The SAC-IA and RANSAC algorithms have an unstable registration performance. The two are prone to being trapped in local optima, which leads to poor registration results. However, the FICP algorithm is not affected by the degree of bucket occlusion and exhibits an excellent performance across the four different poses presented in the figure.

We conducted experiments using 55 frames from the real-world dataset, and Figure 8 shows the registration results between the target point cloud and the real-time point cloud with three different degrees of occlusion. From the figure, it is evident that the red registered point cloud aligns closely with the blue current frame point cloud. The registration performance remains unaffected by variations in bucket occlusion, demonstrating robust registration even under high levels of occlusion. The experimental results show that our method not only has higher accuracy in terms of bucket pose estimation on the simulated dataset but also performs well on the real-world dataset with more interference, such as dust and noise.

Figure 9 shows the registration results when the bucket is heavily occluded in both the simulated dataset and the real-world dataset. In the simulation environment, the bucket is heavily occluded during the excavation process, resulting in significant point cloud loss. In the real-world environment, in addition to the occlusion caused by ores during excavation, the bucket is also occluded due to dust. The results in Figure 9 demonstrate that our proposed method performs well in both the real-world environment and simulation environment, even when the bucket is heavily occluded. Moreover, this method effectively handles occlusions caused by both dust and ores.

The experiments were conducted on a computer with an NVIDIA RTX 4090 GPU and an Intel Core i5-10400F CPU. The average time of bucket pose estimation for our method is 0.095 s. This can effectively meet the practical requirements of bucket pose estimation in terms of both accuracy and computational efficiency.

## 5. Conclusions

In this paper, we propose a method for bucket pose estimation based on 3D vision to solve the large registration error problem caused by occlusion. We collected our own dataset for training and adjusted parameters based on the original PT model, effectively improving the model’s performance and segmentation accuracy (achieving an mIoU metric of 0.9816). We used point cloud Pose Canonicalization to transform the registration between consecutive frames into the registration between the current frame and the first frame’s point cloud. This approach helps to avoid the issue of the low overlap between bucket point clouds in consecutive frames, which can negatively impact registration accuracy. By doing this, we effectively address the challenge of difficult registration when the bucket is occluded. Based on the experiments and analysis presented above, the proposed method achieves the state-of-the-art accuracy (1.2058^◦^, 2.51 cm) in bucket pose estimation, even if the bucket suffers from variable degrees of occlusion, which plays an important role in enhancing the level of intelligence in mining excavators.

## Figures and Tables

**Figure 1 sensors-23-06958-f001:**
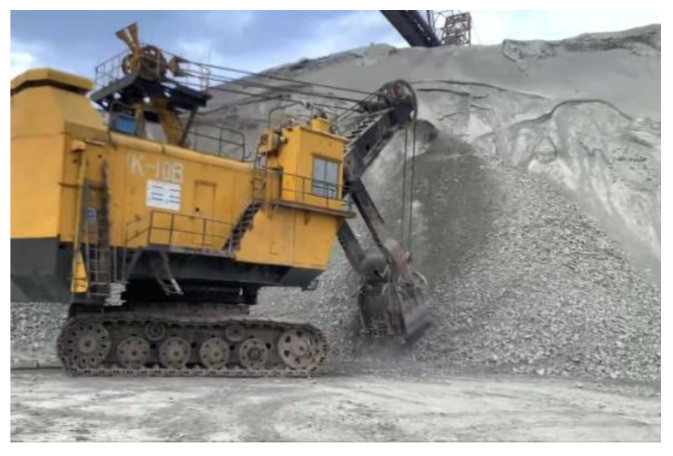
The traditional open-pit mine site.

**Figure 2 sensors-23-06958-f002:**
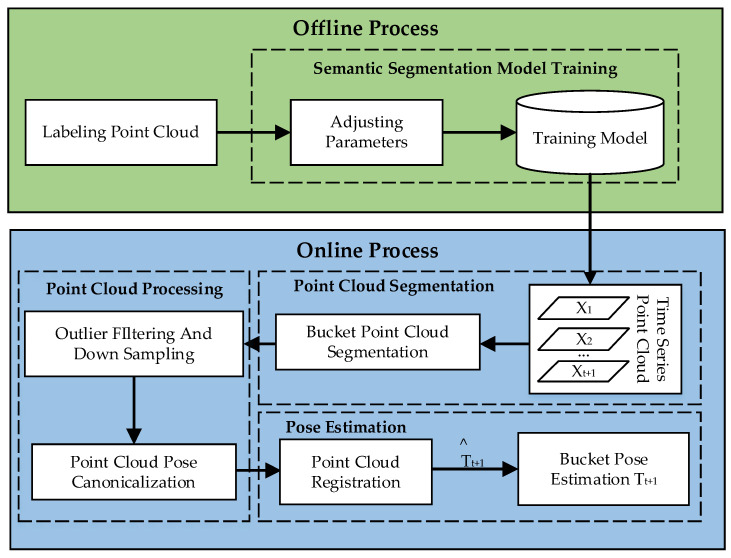
The proposed bucket pose estimation framework.

**Figure 3 sensors-23-06958-f003:**
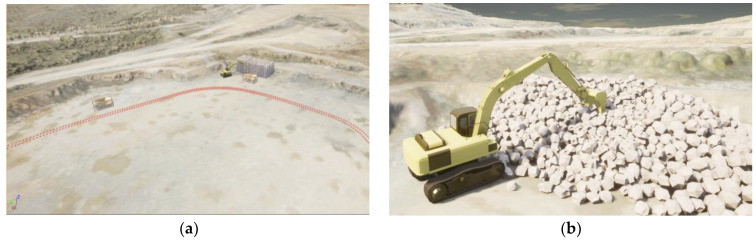
The simulation environment built in CARLA; (**a**) simulate open-pit mine scenarios; (**b**) working scenario of excavator.

**Figure 4 sensors-23-06958-f004:**
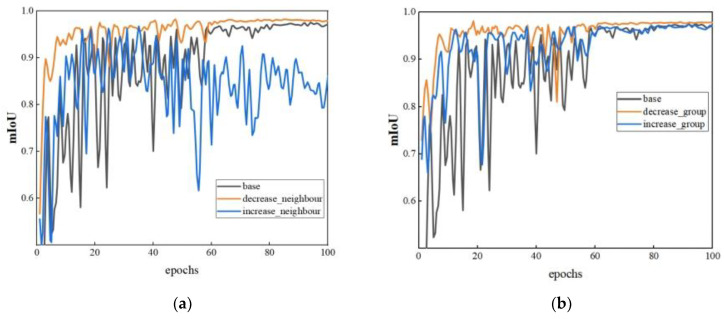
The performance of the PT model when adjusting different parameters. (**a**) Neighboring points; (**b**) the number of groups; (**c**) dropout rate; (**d**) network layers.

**Figure 5 sensors-23-06958-f005:**
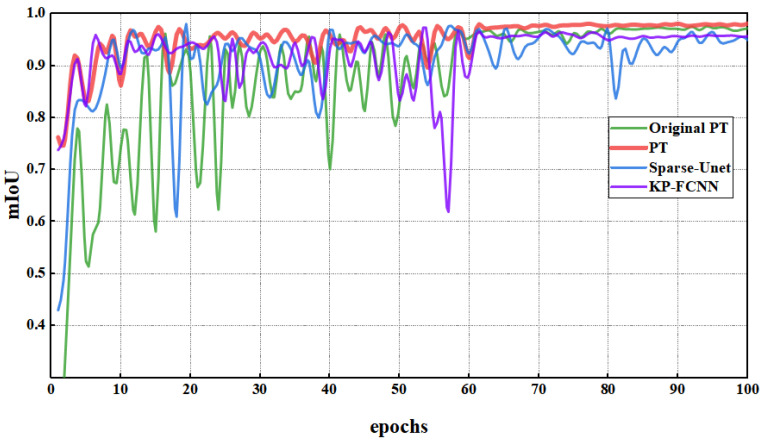
The mIoU curves of different models.

**Figure 6 sensors-23-06958-f006:**
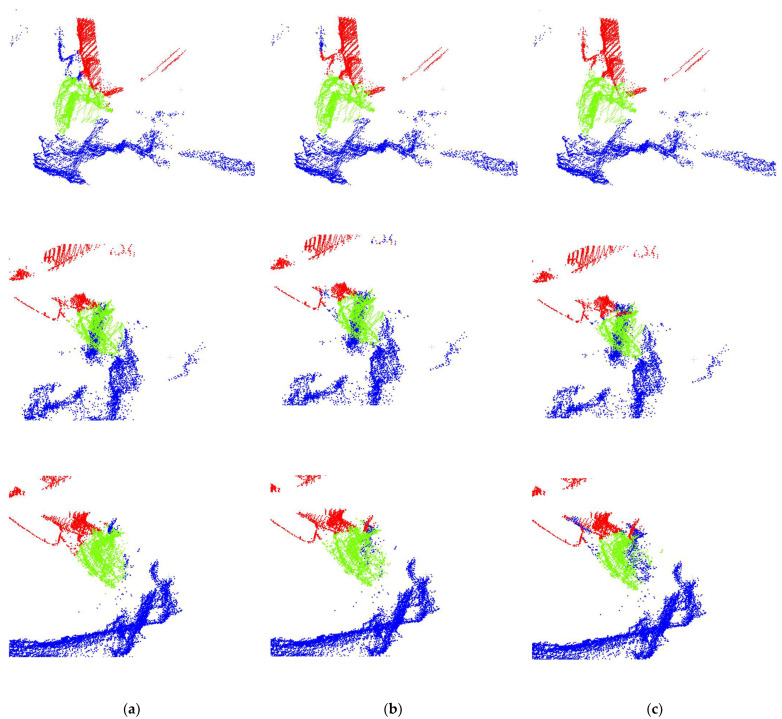
The visualized comparison of the output segmented point clouds. (**a**) Ground truth; (**b**) PT; (**c**) Sparse-Unet.

**Figure 7 sensors-23-06958-f007:**
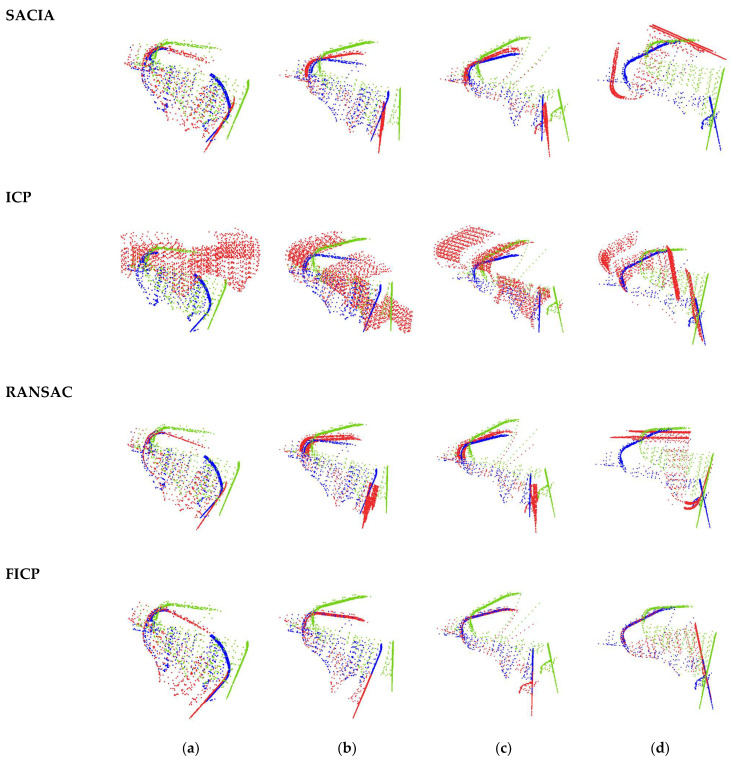
The registration results of the bucket at poses (**a**–**d**) with four registration methods. The target point clouds are marked in blue. The real-time point clouds are marked in green. The transformed point clouds are marked in red.

**Figure 8 sensors-23-06958-f008:**
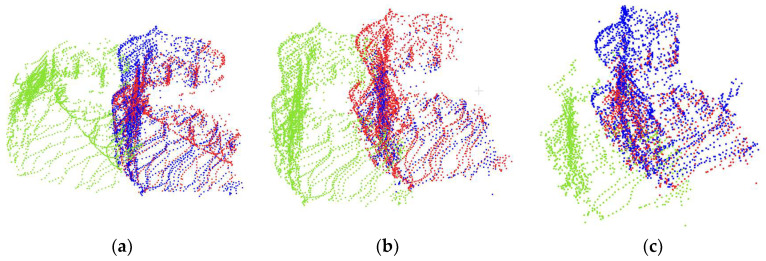
Registration results of bucket at poses (**a**–**c**) for the real-world dataset. The target point clouds are marked in blue. The real-time point clouds are marked in green. The transformed point clouds are marked in red.

**Figure 9 sensors-23-06958-f009:**
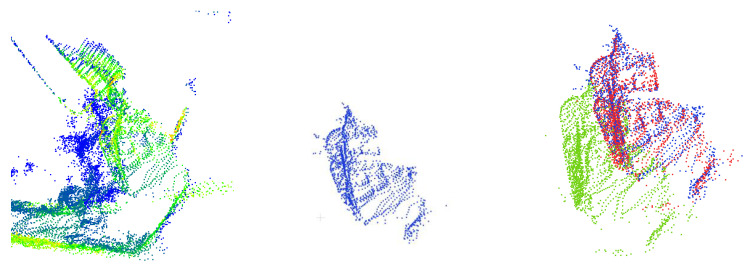
Registration results when the bucket is heavily occluded. (**a**) Original point clouds; (**b**) segmented bucket point clouds; (**c**) registration results.

**Table 1 sensors-23-06958-t001:** Best-performing parameter combination.

Category	Parameters	Original PT	Optimized PT
Model Performance	mIoU	0.9756	0.9813
	Optimal Training Epoch	94	90
Self-Attention Hyperparameters	Neighboring Points	16	8
	Weight-Encoding	GVA	GVA
	Groups	(12, 24, 48)	(6, 12, 24)
Network Hyperparameters	Network Layers	(2, 6, 2)	(2, 6, 2)
	Feature Output Dimensions	(96, 192, 384)	(96, 192, 384)
	Parameter Scale	3.9 million	3.8 million
Training Hyperparameters	Batch Size	6	6
	Dropout Rate	0	0.5

**Table 2 sensors-23-06958-t002:** The PT model parameter adjustments.

Parameters	Base	Neighboring Points		Groups		Dropout Rate		Network Layers	
mIoU	0.9756	0.9677	0.9826	0.9736	0.9796	0.9791	0.9813	0.9691	0.9705
Neighboring Points	16	32	8	16	16	16	16	16	16
Groups	(12, 24, 48)	(12, 24, 48)	(12, 24, 48)	(24, 48, 96)	(6, 12, 24)	(12, 24, 48)	(12, 24, 48)	(12, 24, 48)	(12, 24, 48)
Network Layers	(2, 6, 2)	(2, 6, 2)	(2, 6, 2)	(2, 6, 2)	(2, 6, 2)	(2, 6, 2)	(2, 6, 2)	(3, 8, 3)	(1, 4, 1)
Parameter Scale	3.9 million	3.9 million	3.9 million	4 million	3.8 million	3.9 million	3.9 million	5.34 million	2.47 million
Batch size	6	6	6	6	6	6	6	6	6
Dropout Rate	0	0	0	0	0	0.25	0.5	0	0

**Table 3 sensors-23-06958-t003:** The results for the simulated dataset.

Method	OA	mA	mIoU	Background	Boom	Bucket
Sparse-Unet	0.9932	0.9781	0.9693	0.9924	0.9845	0.9309
OPT	0.9947	0.9863	0.9777	0.9954	0.9849	0.9529
PT	0.9956	0.9873	0.9816	0.9960	0.9882	0.9605

**Table 4 sensors-23-06958-t004:** The results for the real-world dataset.

Method	OA	mA	mIoU	Background	Boom	Bucket
Sparse-Unet	0.9645	0.9433	0.8960	0.9637	0.8953	0.8290
OPT	0.9269	0.8568	0.8012	0.9161	0.8599	0.6276
PT	0.9595	0.9144	0.8853	0.9533	0.8321	0.8703

**Table 5 sensors-23-06958-t005:** Rotation errors (◦) and translation errors (m) of bucket pose estimation.

Method	RE (◦)	TE (m)	Time (ms)
SAC-IA	12.0196	0.2195	725
ICP	20.2733	0.4019	1140
RANSAC	8.9154	0.6853	4386
SIFT+RANSAC	4.0621	0.0559	1011
GICP	2.0621	0.0312	20
FICP	1.2058	0.0251	75

## Data Availability

Data sharing not applicable.

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
