# Peer review of "Real-Time Bucket Pose Estimation Based on Deep Neural Network and Registration Using Onboard 3D Sensor"

_sensors, 2023, doi:10.3390/s23156958_

Round 1
Reviewer 1 Report
Please see the attachment.

English language need to be revised throughout the paper, as there are some grammar issues that need to be fixed.
Reviewer 2 Report
The paper presents the experimental results of the proposed method for bucket pose estimation based on deep neural networks and registration using onboard 3D sensors. The authors collected their dataset for training and testing, and the experimental results are compared with other relevant studies, validating the accuracy and effectiveness of the proposed method. The proposed method achieved better segmentation accuracy and faster registration, enabling real-time pose estimation. However, due to the interference factors in real scenes, such as bucket occlusion and dust near the bucket, the segmentation accuracy of the real scene dataset is slightly lower compared to the simulated environment dataset. Nonetheless, the proposed method still achieves satisfactory segmentation results.
I think the following should be answered or justified:
One potential limitation could be the requirement of an onboard 3D sensor for real-time bucket pose estimation. This may limit the applicability of the proposed method to excavators that do not have an onboard 3D sensor. Additionally, the proposed method may not work well in environments with extreme lighting conditions or where the bucket is heavily occluded. However, these limitations are not discussed in the paper.
Round 2
Reviewer 1 Report
I think the paper deserves publication in the current form, as my previous concerns have been addressed.